



**Demonstrating the "Unit Hydrograph" and flow routing processes**
**involving active student participation – A university lecture experiment**
Karsten Schulz[1], Reinhard Burgholzer[1,2], Daniel Klotz[1], Johannes Wesemann[1], Mathew
Herrnegger[1]
[1]Institute of Water Management, Hydrology and Hydraulic Engineering (IWHW), University of Natural Resources
and Life Sciences (BOKU), Vienna, Austria.
[2]VERBUND Trading GmbH, Am Hof 6a, 1010 Vienna, Austria
Correspondence to: Karsten Schulz (karsten.schulz@boku.ac.at)
**Abstract**
The unit-hydrograph (UH) has been one of the most widely employed hydrological modelling techniques to predict
rainfall-runoff behavior of hydrological catchments, and is still used up-to-date. Its concept is based on the idea
that a unit of effective precipitation per time unit (e.g. mmh$^{-1}$) will always lead to a specific catchment response in
runoff. Given its relevance, the UH is an important topic addressed in most of the (engineering) hydrology courses
at all academic levels. While the principles of the UH seem to be simple and easy to understand, teaching
experiences in the past suggest strong difficulties in students' perception of the UH theory and application.
In order to facilitate a deeper students' understanding of the theory and application of the UH, we developed a
simple and cheap lecture theatre experiment involving an active student participation. The seating of the students
in the lecture theatre represented the "hydrological catchment" in its size and form. A set of plastic balls, prepared
with a piece of magnetic strip to be tacked to any white/black board, each represented a unit amount of effective
precipitation. The balls are evenly distributed over the lecture theatre and routed by some given rules down the
catchment to the "catchment outlet", where the resulting hydrograph is monitored and illustrated at the black/white
board.
The experiment allowed an illustration of the underlying principles of the UH, including stationarity, linearity and
superposition of the generated runoff and subsequent routing. In addition, some variations of the experimental
setup extended the UH-concept to demonstrate the impact of elevation, different runoff regimes and non-uniform
precipitation events on the resulting hydrograph.
In summary, our own experience in the classroom, a first set of student exams, as well as student feedback and
formal evaluation suggest that the integration of such an experiment deepened the learning experience by active
participation. The experiment also initialized a more experienced based discussion of the theory and assumptions
behind the UH.  Finally, the experiment was a welcome break within a 3-hour lecture setting, and great fun to
prepare and run.





**1    Introduction**
1.1    *Background*
The prediction of catchment rainfall-runoff behavior is an important prerequisite of any effective flood risk and
water resources management practice. Rainfall-runoff modelling has a long history with a starting point that can
been dated back more than 150 years to the work of Mulvaney (1851) and even further (see e.g. Biswas et al.,
1970). He first introduced a simple linear relationship between peak discharge and maximum catchment average
rainfall intensity that is dependent on catchment size and an empirical coefficient that effectively represents all
other catchment characteristics (Beven, 2012). It is known as the *rational method* in engineering hydrology, and
with modification is still used today (e.g. Hromadka, 1994;  Plate, 1988).
A step further has been the first attempt of a spatially distributed hydrological model by Ross (1921). In his
approach, the catchment was split up into zones of equal travel times to the catchment outlet, and runoff production
is calculated for each area, dependent on the antecedent conditions and rainfall rates. The resulting time-area
diagrams represent the delays for runoff from each part of the catchment. Similar concepts have been introduced
by e.g. Zoch (1934) or Clark (1945) and are still included in current distributed hydrological model systems.
While it has been long known that flow velocities change in a nonlinear way with flow rate or flow depth, the
Ross-approach relies on the assumption of linearity in routing the runoff to the catchment outlet. This violation
however, has been shown less critical compared to the problem of estimating the effective rainfall for each event
{Beven, 2012)}. The effective precipitation is thereby the part of the total precipitation contributing to runoff
during an event. The estimation of this proportion is generally a nonlinear process depending on the antecedent
catchment conditions, including soil moisture and interception storage conditions, possible snow cover, as well as
rainfall intensities. Common approaches to calculate effective precipitation are the *constant loss rate* (φ-index)
method, the *constant proportion* method, or infiltration based methods as suggested by Horton (1933).
Another major difficulty with the Ross-approach was to decide which areas of the catchment would contribute to
the different time zones, since there were almost no information on flow velocities and pathways in the different
soil compartments (surface runoff, interflow) and on different conditions available. The unit-hydrograph (UH)
method developed by Sherman (1932) tried to avoid these difficulties by representing the various time delays for
runoff generated within the catchment by a stationary time distribution that has not necessarily any direct link to
a particular location. The principle idea of the method is that assuming a linear routing procedure, this distribution
could be normalized to represent the response to a unit of runoff production, or effective rainfall, generated over
the catchment in one time step. In other word, the UH represents a discrete transfer function for effective rainfall
to reach the basin outlet, lumped to the scale of the catchment. The UH has been one of the most widely employed
hydrological modelling techniques to predict rainfall-runoff behavior of hydrological catchments, and is still used
in current generations of hydrological forecasting systems {see e.g. Samaniego, 2010; Michel, 2003).
Given the importance of the UH in the historical development of rainfall-runoff models, and the fact that its basic
principles are still included in current spatially distributed hydrological models, the UH is an important subject in
any of the academic hydrology courses at the B.Sc. and M.Sc. level. The UH theory and principles include general





concepts of catchment hydrology that form the foundation for further, more advanced topics and therefore need to
be introduced in an understandable way.
*1.2 Teaching Situation*
The principles of runoff generation processes and the basic concepts for rainfall-runoff modelling are essential
parts of the 3rd semester "Hydrology and Water Resources Management 1" course that is mandatory within the
"Civil Engineering and Water Management (H033231)" program at the University of Natural Resources and Life
Sciences (BOKU), Vienna. All students in the program have previously attended introductory courses in
mathematics, physics, statistics, mechanics, hydraulics, and went through an introduction into hydrography and
hydrometry. While in theory the previous knowledge seems to be more than sufficient and adequate to understand
and grasp the central ideas behind the UH, exam results over the last years consistently and repeatedly
demonstrated significant gaps in students' understanding. While the concept of "effective precipitation"
contributing to catchment runoff did not seem to make any difficulties, it was in particular the interpretation of the
UH within a hydrological system context and its use as a prediction tool that were identified as critical areas.
In addition to the standard slide presentation of the theory, we decided in the winter-semester 2016/17 to
additionally visualize the concept of the unit-hydrograph, its underlying assumptions and the linear routing
principles in a lecture theatre experiment. In our experiment, students and the location they were sitting represented
a hydrological catchment in the lecture theatre, and they were actively involved in routing an effective precipitation
event to a fictive runoff gauge in the room. This gauge was "monitored" over time and the resulting runoff data
was visualized in an illustrative way.
The setup of this experiment, the material required and its preparation as well as variants of the experiments and
possible follow-up discussions to interpret these experiments will be described in some detail in the following.
**2   Experimental Setup**
2.1   The Unit Hydrograph (UH)
Summarizing the theory of the UH as briefly described in the previous section, three basic principles can be
emphasized when transferring a unit of effective precipitation to the catchment outlet. These are: i) stationarity, ii)
linearity, and iii) superposition. Stationarity of the UH concept assumes that a unit of effective precipitation will
always generate the same hydrograph, independent of the time of the day/year and of the antecedent e.g. soil
moisture conditions. Linearity in the routing process requires that any event with x units of effective precipitation
will generate x-times as much predicted runoff in the hydrograph at the catchment outlet, having the same temporal
distribution. The principle of superposition states that multiple consecutive effective precipitation events can be
treated independently and predicted runoff in the hydrographs is superimposed by simply adding individual
responses. An example unit hydrograph and these three fundamental principles are illustrated in Fig. 1.




We would like to point here, that the UH concept per se does not explicitly link the runoff generation processes to
any particular location. The UH rather represents a travel time distribution of a unit effective rainfall that is
homogeneously distributed over the catchment. Being aware of this fact, we nevertheless chose a spatial explicit
setting for illustrating the UH and its principles as well as runoff routing processes. The implementation into a
lecture theatre experiment will be explained step-by-step in the next subsection.

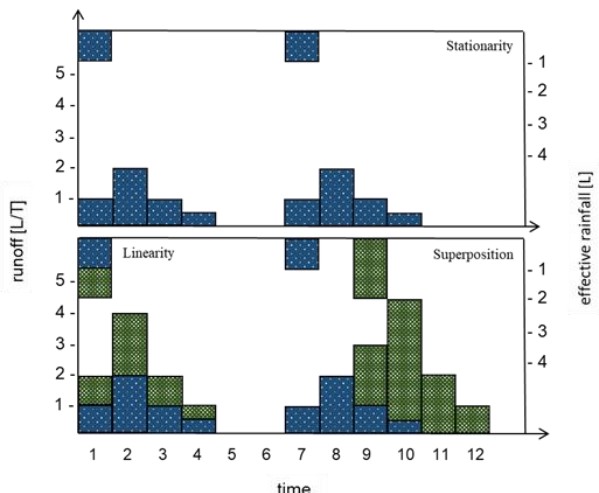

Figure 1: An example of a UH (upper left) and an illustration of the principles of stationarity (upper right), linearity (lower
left), and superposition (lower right). Units for runoff are given normalized by the catchment area in length [L] per time [T].
2.2    Lecture theatre experiment and teaching material
The following steps were implemented to realize an experimental illustration of the UH principle in a lecture
theatre. The experiment took about 25 min including all instructions, explanations and some basic discussions. It
will be shown later (section 2.3) that different variants of the experiments can be performed to focus on further
aspects of catchment rainfall-runoff behaviour so that the experiment including discussions can be easily extended
to 90min. The experiment uses very simple and cheap materials that were easily available in any department store.
Overall costs were in the range of €30,-.
Step 1 (ball preparation):
An effective precipitation of 1mm is realized using differently colored plastic balls. A short piece of magnetic
stripe was glued to each ball so that they could be tacked to most of the lecture theatre and seminar room
white/black boards. Figure 2 illustrates a prepared ball and its use at the white/black board.
Step 2 (defining the catchment):
The "size" and the "form" of the catchment is defined by the positioning/seating of the individual students in the
lecture theatre. By considering a different number of students within the experiment and changing the seating
positions, the catchment size and form can be varied in order to examine and illustrate the effect of variations in
both on the resulting hydrograph. Figure 3 shows the lecture theatre setup – each student holding a plastic ball
representing 1mm of effective precipitation received by the catchment.
One person - approximately in the center of the first row or a teaching assistant (see Fig.3, left, gauge) - is defined
as the catchment outlet, thus receives all the effective precipitation (balls) that have been routed "down" the
catchment (lecture theatre).

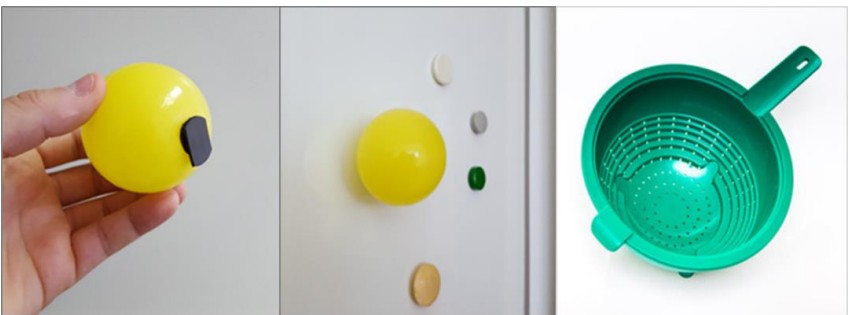

Figure 2: A piece of magnetic stripe glued to a plastic ball (left) that represents a unit amount (1mm) of effective rainfall, as
well as generated specific runoff. It can be tacked to most of the classroom white/black boards (middle). The sieve is used to
"collect" generated runoff during the routing procedure (right).

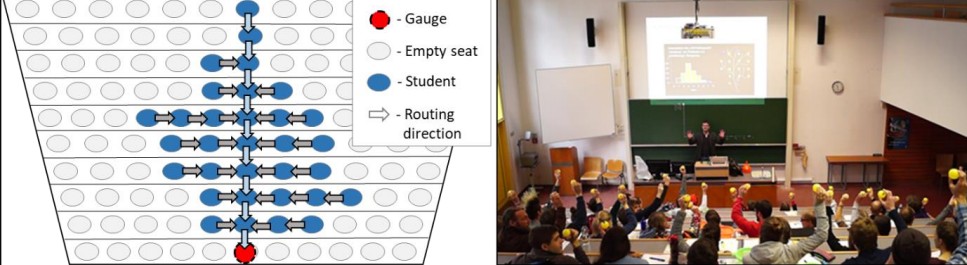

Figure 3: Catchment setup by 32 students in the original lecture theatre (right). The catchment/each student received an effective
precipitation of 1mm represented by a plastic ball. A sketch of student seating, flow routing and the gauge (left). Blue flow
arrows indicate the main stream channel within the catchment.
Step 3 (routing rules):
The routing of the effective precipitation within the catchment has to follow some rules in order to generate runoff.
To mimic some real catchment behavior, each student has to transport her/his 1mm of effective precipitation water
package towards the outlet. A simple routing scheme that is easy to explain and to execute is sketched in Fig. 3
(left) and proceeds as follows for each time step: i) vertical transport (blue arrows) of water packages of one
position only along the main stream channel starting in the first row; the water packages in the gauge position are
the measured runoff information for this time step; ii) horizontal transport (grey arrows) of one position towards
the main stream channel in each row starting with the inside positions. The sampling of the water packages (balls)





and along is carried out with plastic sieves as the number of balls might exceed the number of balls that can be
handled by hand (see Fig. 2, right).
Step 4 (hydrograph representation):
With each time step, a different number of water packages (balls) are collected/sampled at the catchment gauge.
They represent the catchment hydrograph resulting from 1mm of effective precipitation homogeneously spread
over the entire catchment. Given the routing scheme that has been chosen for this particular experiment, the
hydrograph in Fig. 4 represents the corresponding UH of our lecture theatre catchment.  The units of the y-axis are
in length per time and dependent on the units chosen for the time steps and the effective precipitation (here 1 mm
per ball).

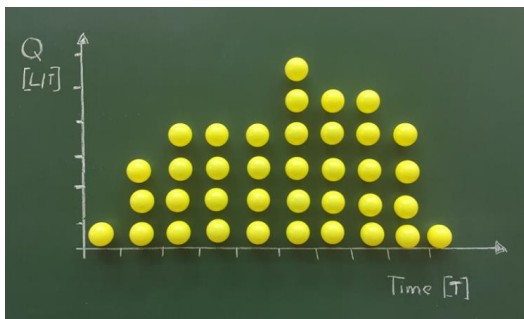

Figure 4: A lecture theatre catchment unit-hydrograph resulting from 1mm effective precipitation (represented by a yellow
plastic ball) homogeneously spread over the catchment.
**3    Results and Discussion**
The UH illustrated in Fig. 4 is the result of a specific lecture theatre setting including the form of the catchment
defined by the positioning of the students in the lecture theatre (see Fig. 3) and the rules formulated to route the
effective precipitation through the catchment. The basic experimental setup as described in section 2 can be
extended in various ways to illustrate i) the basic principles behind the unit-hydrograph, and ii) a number of
additional factors that have a significant control on the generated hydrograph, such as the catchment terrain, the
surface conditions and dominant runoff components, for example surface runoff or subsurface stormflow.
3.1    Illustration of unit hydrographs principles
The three principle underlying the UH are stationarity, linearity and superposition of runoff generated by different
events (Fig. 1). The principle of stationarity is easily explained by simple repeating the experiment and thereby
receiving the same hydrograph as result. Such a repetition might be not very exciting for the students, but
experience shows us that a thought experiment is sufficient to foster understanding. The principle of linearity could
be illustrated and discussed in two ways. First, given the initial experiment where each plastic ball represented a
unit amount of effective precipitation (e.g. 1mm), this is simply redefined to twice the amount (e.g. 2mm) which
exactly doubles the runoff generated in the hydrograph. Secondly, the experiment may be repeated with twice the




number of balls per student, again resulting in a doubling of the runoff in the generated hydrograph. The second
variant will obviously provide a more vivid illustration of the principle of linearity; however, it will require more
resources in terms of number of balls and more experimental execution time, which might be more limiting given
any lecture setting. The principle of superposition can be illustrated by considering two independent precipitation
events, each producing e.g. 1mm of effective precipitation. The different precipitation events can be labeled by
using differently colored balls (e.g. yellow and blue). While the first precipitation event will take place at the first
time step, the second event will have a time delay of some steps. Figure 5 (right) shows the superposition of the
two events (each 1mm of effective rainfall) and the resulting hydrograph.
As mentioned earlier, the unit hydrograph does not explicitly link any runoff generation to a specific location in
the catchment. We here used an explicit setting to illustrate the principles of the UH – each explicit setting or form
of a catchment and routing scheme will produce a specific hydrograph. The experiment in Fig. 5 provides a good
opportunity to discuss this fact with the students. This point could be illustrated by designing a second but different
experimental setting (form and/or routing scheme) that leads to the same hydrograph.

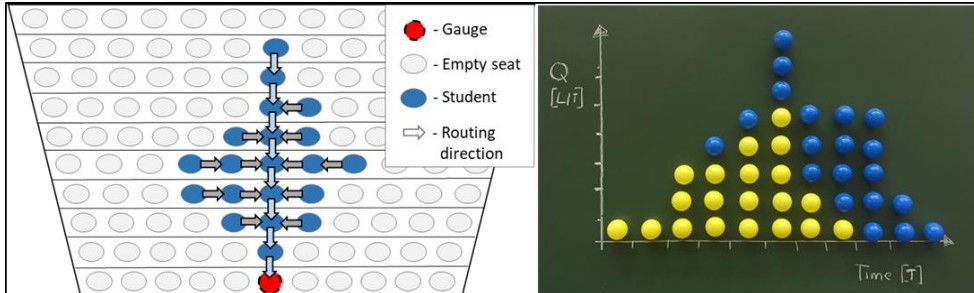

Figure 5: The superposition of two rainfall events, both producing an effective precipitation of 1mm represented by a yellow
ball (first event) and a blue ball (second event, delayed by 4 time steps), and the resulting hydrograph (right). The left figure
illustrates the student seating and routing scheme applied in this experiment.
**3.2    Analysis of factors controlling the unit-hydrograph**
One obvious factor to analyze is the form of the catchment and its impact on the dynamics of the hydrograph.
While such an analysis is given in many of the recent hydrology textbooks {e.g. \Baumgartner, 1996 #8160`,
p.525f; \Ward, 2003 #8161`, p.126f}, this effect is simply assessed by changing the seating positions of the
students in the lecture theatre and repeating the experiment. Figure 6 illustrates the representation of different
catchment forms by student seating in the lecturer theatre (top panel) and the resulting hydrographs (lower panel).
While the concept of the UH per-se does not require any linkage between the time delay of any runoff generated
in the catchment to a particular location (see section 1), the lecture theatre experiment is well suited to additionally
discuss any of the catchment properties that are related to the different forms of runoff processes in the catchment.
In the standard routing scheme as introduced in section 2, no effect of differences in topography is considered in
the routing scheme. In order to investigate the effects of topography, different areas with steep elevation gradients
(e.g. steep areas close to the catchment boundaries, where the transport of water is enhanced by allowing balls to
"jump" over 2 positions, thus representing the increase in flow velocities due to increased potential gradients.


Differences in the soil texture and resulting infiltration capacities might be represented similarly, indicating areas
with more frequent surface runoff generation and quick flow conditions versus areas with more dominant
subsurface or interflow flow regimes that deliver water more slowly towards the catchment outlet. In this way,
different routing rules can be linked to different catchment properties and runoff/flow regimes, which can be used
for critical discussion of the assumptions and underlying principles of the UH, as well as the possible effects in
case those principles and assumptions are violated.

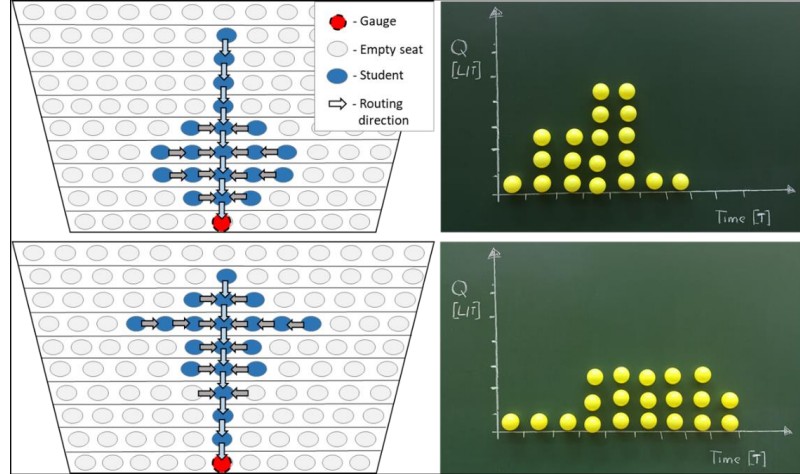

Figure 6: The effect of different catchment forms (same area/number of students, left) on the resulting hydrographs (right).
While beyond the underlying assumptions of the UH (spatially uniform effective precipitation), this lecture theatre
experiment is well suited to additionally demonstrate the effect of spatially non-uniform precipitation and runoff
generation processes, by limiting the distribution of plastic balls to only a subset of the students. Figure 7 illustrates
the resulting hydrographs when only the lower half (left) or the upper half (right) of the catchment receive a unit
amount of effective precipitation. The differences in the timing of the resulting hydrographs are clearly visible.
**4    Experience and Outlook**
Our personal experience with the unit-hydrograph and runoff routing lecture theatre experiment, as well as the
direct student feedback and the student evaluation was extremely positive and revealed the following key aspects:
(1)   The experiment was well suited to illustrate the concepts of the unit-hydrograph. The underlying

18         principles such as stationarity, linearity and superposition, as well as the effect of different catchment

19         form, catchment elevation, runoff mechanism and non-uniform precipitation events on the catchment

20         hydrograph could be well demonstrated.

(2)   The active participation of every student in the experiment allowed an intensive "active experience" of

22         the UH and routing mechanism by being "one part of the catchment" and being "involved" in the transport

23         of the water towards the catchment outlet. Student's active participation in the experiment also highly

24         supported an intensive discussion on the theoretical background of the unit-hydrograph. Assumptions

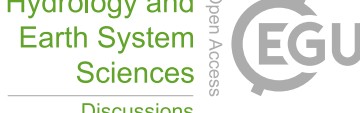



such as the linearity of routing procedure could be questioned and discussed with regard to real catchment
behaviour including the impact of antecedent soil moisture conditions on the runoff generation, and the
non-linearity of flow depth – flow velocity relationship.
(3) Depending on the number of experiments conducted, the time needed for a single experiment including
the discussion will range between about 30 – 90 min. Given a 3-hour framework for our particular lecture,
the experiment was a very welcome "active break" from the standard slide based lecturing. It refocussed
students' minds and attention.
(4) Actively participating in deriving the catchment rainfall-runoff relationship was fun, not only for the
students, but also for everybody involved in the course teaching.
To summarize our experience with the integration of such a lecture theatre experiment into the course: We believe
that overall a much deeper learning experience for the students could be achieved due to the visualization and
active participation of the students. This experience is well expressed by a statement of Confuzius (551-479 BC)
saying that "I hear and I forget, I see and I remember, I do and I understand" and will encourage us to further
extend the integration of experiments into lecture-based teaching.

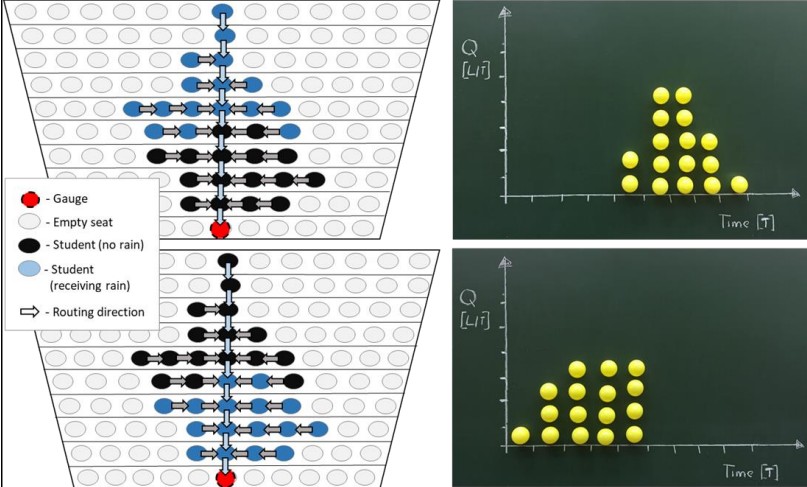

Figure 7: The effect of different spatial distributions of effective precipitation within the catchment on the resulting hydrograph.
The student seating, the spatial distribution of effective precipitation and the routing scheme are indicated on the left, the
hydrograph are illustrated on the right.
*Data Availability.* All required information and appropriate references are given in the text.
*Author contributions.* KS (PI) designed the experiment and run the lecture. RB (M.Sc. student) developed the
teaching material. JW (PhD student), DK (PhD student), MH (Co-PI) as well as RB contributed to the development
of the teaching experiment. All authors were involved in the manuscript and figure preparation.
*Competing interest.* The authors declare that they have no conflict of interest.





*Acknowledgement.* The authors are thankful to the group of students in the 3rd semester "Hydrology and Water
Resources Management 1" course within the "Civil Engineering and Water Management (H033231)" program at
the University of Natural Resources and Life Sciences (BOKU), Vienna.
**Literature**
Baumgartner, A., and Liebscher, H. J.: Lehrbuch der Hydrologie, Band 1 Allgemeine Hydrologie - Quantitative
Hydrologie, 2. ed., Gebrüder Bornträger, Berlin, Stuttgart, 1996.
Beven, K.: Rainfall-Runoff Modelling: The Primer, 2nd ed., J. Wiley & Sons, Ltd., 2012.
Biswas, A. K.: History of hydrology North-Holland Publishing Company, Amsterdam-London, 1970.
Clark, C. O.: Storage and hydrograph, T Am Soc Civ Eng, 110, 1416-1446, 1945.
Horton, R. E.: The role of infiltration in the hydrologic cycle, Transactions-American Geophysical Union, 14, 446-
11    460, 1933.

Hromadka, T. V., and Whitley, R. J.: The Rational Method for Peak Flow-Rate Estimation, Water Resour. Bull.,
13    30, 1001-1009, 1994.

Michel, C., Perrin, C., and Andreassian, V.: The exponential store: a correct formulation for rainfall-runoff
modelling, Hydrological Sciences Journal-Journal Des Sciences Hydrologiques, 48, 109-124,
doi:10.1623/hysj.48.1.109.43484, 2003.
Mulvaney, T. J.: On the use of self-registering rain and flood gauges in making observations of the relations of
rainfall and flood discharges in a given catchment, Proceedings of the Institution of Civil Engineers of Ireland, 4,
19    19-31, 1851.

Plate, E. J., Ihringer, J., and Lutz, W.: Operational Models for Flood Calculations, J. Hydrol., 100, 489-506,
doi:10.1016/0022-1694(88)90198-9, 1988.
Ross, C. N.: The calculation of flood discharge by the use of time contour plan isichrones, Transactions of the
Institute of Engineers, Australia 2, 85-92, 1921.
Samaniego, L., Kumar, R., and Attinger, S.: Multiscale parameter regionalization of a grid-based hydrologic model
at the mesoscale, Water Resour. Res., 46, doi:10.1029/2008wr007327, 2010.
Sherman, L. K.: Streamflow from rainfall by unit-graph method, Engineering News Record, 108, 501-505, 1932.
Ward, A. D., and Trimble, S. W.: Environmental Hydrology, 2. ed., Lewis Publishers, New York, 2003.
Zoch, R. T.: On the relationship between rainfall and streamflow, Mon. Weather Rev., 62, 315-322, 1934.