# Peer review of "Demonstrating the "Unit Hydrograph" and flow routing processes"

_Hydrology and Earth System Sciences, 2017_

## Referee Comment (RC1) · Anonymous Referee #1 · 15 Sep 2017

This paper presents a method for interactively teaching students about the unit hydrograph. The approach taken is simple, involving the students passing balls along defined flow pathways so that the result at the "catchment outlet" can be observed. It is a simple, low-cost method of demonstrating a simple case of "unit hydrograph". Given the time needed to run each "experiment", I feel that a hybrid approach would be better, where the idea is introduced using a simple participatory demonstration as described here, but more detailed experiments are done through computer simulation. This is particularly the case when the time needed for a single experiment (including discussion) is between 30 and 90 minutes (page 9, line 4-5). 90 minutes is a considerable break in a 3 hour lecture, and suggests a more efficient method might be needed.

The real question here is: how many such experiments are needed in order to provide a suitable improvement in student understanding? Can a combination of participatory and computer examples achieve the same effect in less time?

The paper gives a reasonable review of the history of the unit hydrograph. I consider that the authors are incorrect in saying that the effective rainfall is homogeneously distributed over the catchment (page 4, lines 2-3). This is not necessarily the case. What the UH concept considers is that the spatial distribution of effective rainfall doesn't change between events. It can be non-homogeneously distributed. This can be due to spatial variations in rainfall (e.g. due to topographic effects), or due to spatial variations in the fraction of rainfall that is converted into effective rainfall (e.g. due to topography, soils, vegetation). Considering the effective rainfall to be homogeneously distributed across the catchment is a simple case, but not really the requirement of the unit hydrograph concept.

I think papers like this do have a place in HESS - but this paper needs a little more work in order to be of publishable quality.

---

## Referee Comment (RC2) · Anonymous Referee #2 · 2 Jan 2018

1. This paper presents a creative method of teaching the Unit Hydrograph, a fundamental concept in hydrology, in an interactive and non-lecture format. This would be a good addition to HESS as more and more instructors hope to incorporate non-traditional and active methods of teaching STEM concepts to suit different types of learning styles. While the demonstration may not be feasible in some cases, this paper presents one option of teaching the UH concept and could be the basis of different modifications to suit individual classroom needs.

2. One general concern I had was that the authors cited "strong difficulties in students' perceptions of the UH' as the motivation for using the active demonstration. However,

[Figure]

I could not find what specifically the previous concerns were and if they were actually addressed/reflected in the final evaluations after the demonstration. I believe summarizing some of these learning difficulties and how the demonstration overcomes them would help convince other readers to try this method, especially if they are encountering the same issues with their students.

3. The organization of the paper as well as the figures are of good quality; the only concern regarding the writing pertains to some awkward phrasing and some typographic errors (see technical corrections in supplement).

Please also note the supplement to this comment:
https://www.hydrol-earth-syst-sci-discuss.net/hess-2017-498/hess-2017-498-RC2-supplement.pdf

─────────────────────────────

[Figure]

**Supplement:**

**General Comments**

1. This paper presents a creative method of teaching a fundamental concept in hydrology in an interactive and non-lecture format. This would be a good addition to HESS as more and more instructors hope to incorporate non-traditional methods of teaching STEM concepts to suit different types of learning styles. While the demonstration may not be feasible in some cases, this paper presents one option of teaching the unit hydrograph concept and could be the basis of different modifications to suit individual classroom needs.

2. One general concern I had was that the authors cited "strong difficulties in students' perceptions of the UH' as the motivation for using the active demonstration. However, I could not find what specifically the previous concerns were and if they were actually addressed/reflected in the final evaluations after the demonstration. I believe summarizing some of these learning difficulties and how the demonstration overcomes them would help convince other readers to try this method, especially if they are encountering the same issues with their students.

3. The organization of the paper as well as the figures are of good quality; the only concern regarding the writing pertains to some awkward phrasing and some typographic errors (see technical corrections below).

**Specific Comments**

P2, Line 14 – This section starts at a nice review for the UH; however the Zoch and Clark references here are inserted without much description and are vague. If you want to use them as using 'similar concepts', I would suggest you provide more details.

P2, Line 19-21 – You should cite what you say here

P2, Line 25 – Different conditions such as?

P2, Line 33 – I believe that the MHM model in the Samaniego paper does *not* explicitly use the UH concept as a routing method as it summarizes different grid cells through the regionalization process, then upscales to larger spatial scales and is not necessarily constrained to the UH assumptions/limitations. Perhaps double check the use of this citation

P3, Line 12-13 – Could you elaborate a bit more here? Understanding how the students struggle here would help the reader understand how the activity improves their understanding

Figure 1 – The figure illustrates the concept fairly well. I would choose colours/patterns that contrast more. Also make sure that the final version does not have blurry text

**Technical Corrections**

P1, Line 12 – Unit-hydrograph and unit hydrograph are used interchangeably throughout the text and title. You should pick one and be consistent with it

P1, Line 13 – 'up-to-date' is awkward, perhaps use 'to date' or 'to this day'

P1, Line 15 – topic addressed in most  (engineering) hydrology…

P1, Line 19 - experiment involving  active student….

P2, Line 10 – **A step further has been the first attempt** of a spatially distributed…First section is awkwardly phrased.

P2, Line 18 – Check that your references don't have the brackets {} for next submission. They also appear later on.

P2, Line 28 – **Principal** idea, not principle

P2, Line 30 – In other word**s,**

P2, Line 36 -in any  academic hydrology courses at the (BSc and MSc); I would say undergraduate and graduate level as they can be different in institutions or countries, e.g. BASc, B.Eng., etc.

P4, Line 1 – We would like to point **out**  that the UH…

P4, Line 3 – a **spatially** explicit

P4, Line 8 - Units for runoff are  normalized by …

P4, Line 14 – 90_min

P5, Line 21 – The sampling of the water packages  carried out with…

P7, Line 16 - …by a yellow ball (first event from **Figure 1**)

P7, Line 20 - many **of the** recent

P7, Line 29 – (e.g. steep areas close to …. You need to close the parenthesis somewhere

P9, Line 12 – **Confucius**

---

## Author Comment (AC1) · 22 Jan 2018

**Responses to the Reviewer#1 Comments**

(Referee comments in black; Responses in blue)

First, we would like to thank the reviewer for his/her fair and very valuable comments. In the following, we have addressed each reviewer comments in detail and have indicated how we might alter and update the manuscript given the comments. We hope that we have addressed all comments sufficiently, and we are looking forward to your feedback and your decision.

**Comments by Referee #1:**

1. *This paper presents a method for interactively teaching students about the unit hydrograph. The approach taken is simple, involving the students passing balls along defined flow pathways so that the result at the "catchment outlet" can be observed. It is a simple, low-cost method of demonstrating a simple case of "unit hydrograph".*
   We thank the referee for this statement, as it was exactly our intension to develop such a simple, easy to implement and cheap experiment suitable for demonstration within a lecture.

2. *Given the time needed to run each "experiment", I feel that a hybrid approach would be better, where the idea is introduced using a simple participatory demonstration as described here, but more detailed experiments are done through computer simulation. This is particularly the case when the time needed for a single experiment (including discussion) is between 30 and 90 minutes (page 9, line 4-5). 90 minutes is a considerable break in a 3 hour lecture, and suggests a more efficient method might be needed.*
   We fully agree with the reviewer that it is also necessary to have additional exercises (e.g. in the computer lab), where students do explicit calculations applying the unit-hydrograph concept. We also do that in our program within the course "Exercises in Hydrology". The course is conducted separately, but organized in close cooperation. The lecture theatre experiment (as introduced in the paper) is a visceral aid that gives an additional visualization and participatory demonstration. The main goal is therefore to stimulate student interests and to help them in their scientific learning. The positive effect of experimental demonstrations on deeper understanding and learning has been found by a number of authors (e.g. Roberts et al., 2005, Savec et al., 2005) and → we will include these references in a revised version of the manuscript.
   Concerning the time of the experiment – we wrote 30-90min in the submitted manuscript –, we could repeat the experiment and, with the help of 2 student assistants, we were able to include the basic demonstration within a 15-20min time slot. When this time slot is well set, it is an ideal interruption of a 3h lecture.
   In the manuscript, we additionally describe variants of the experiment that can, but do not need to be performed. These variants are only described in case there is more time available. →We will make these points clearer in the revised version of the manuscript.

3. *The real question here is: how many such experiments are needed in order to provide a suitable improvement in student understanding? Can a combination of participatory and computer examples achieve the same effect in less time?*
   This is a very interesting question and depending on some funding, there might be a good chance over the next years to tackle this question. In general, a longer-term educational experiment would be required to answer this question thoroughly. The experiment will use some kind of split group approach and then analyze the exam/learning results for these different settings. As we have received the teaching award of our university for this experiment in the last year, our educational department had approached us in order to discuss such a long-term study to examine the effect. We apologize to admit that currently an answer to this question is out of scope. → However, we suggest briefly outlining and discussing such a longer-term investigation in the discussion.

4. *The paper gives a reasonable review of the history of the unit hydrograph. I consider that the authors are incorrect in saying that the effective rainfall is homogeneously distributed over the catchment (page 4, lines 2-3). This is not necessarily the case. What the UH concept considers is that the spatial distribution of effective rainfall doesn't change between events. It can be non-homogeneously distributed. This can be due to spatial variations in rainfall (e.g. due to topographic effects), or due to spatial variations in the fraction of rainfall that is converted into effective rainfall (e.g. due to topography, soils, vegetation). Considering the effective rainfall to be homogeneously distributed across the catchment is a simple case, but not really the requirement of the unit hydrograph concept.*
   We fully agree on this comment. The assumption of a uniform "effective precipitation" is very often made in many textbooks as a requirement (e.g. Maniak, 2016, p350). → However, we will modify and correct this part of the introduction.

5. *I think papers like this do have a place in HESS - but this paper needs a little more work in order to be of publishable quality.*
   We hope our comments and suggested adaptations will sufficiently address the comments made by Referee #1.

Literature:

Maniak, U.: Hydrologie und Wasserwirtschaft, eine Einführung für Ingenieure, 7.Aufl., Springer Vieweg, 2016

Roberts, J. R., Hagedorn, E., Dillenburg, P., Patrick, M., and Herman,T.: Physical models enhance molecular three-dimensional literacy in an introductory biochemistry course, Biochem. Mol. Biol. Edu., 33, 105–110, 2005.

Savec, V. F., Vrtacnik, M., and Gilbert, J. K.: Evaluating the educational value of molecular structure representations, in: Visualisation in Science Education, edited by: Gilbert, J. K., Springer, 269–300, 2005.

---

## Author Comment (AC2) · 22 Jan 2018

**Responses to the Reviewer #2 Comments**

(Referee comments in black; Responses in blue)

First, we would like to thank the reviewer for his/her fair and very valuable comments. In the following, we have addressed each reviewer comments in detail and have indicated how we might alter and update the manuscript given the comments. We hope that we have addressed all comments sufficiently, and we are looking forward to your feedback and your decision.

**Comments by Referee #2**

1. *This paper presents a creative method of teaching the Unit Hydrograph, a fundamental concept in hydrology, in an interactive and non-lecture format. This would be a good addition to HESS as more and more instructors hope to incorporate nontraditional and active methods of teaching STEM concepts to suit different types of learning styles. While the demonstration may not be feasible in some cases, this paper presents one option of teaching the UH concept and could be the basis of different modifications to suit individual classroom needs.*
   We thank the referee for this statement!

2. *One general concern I had was that the authors cited "strong difficulties in students' perceptions of the UH' as the motivation for using the active demonstration. However, I could not find what specifically the previous concerns were and if they were actually addressed/reflected in the final evaluations after the demonstration. I believe summarizing some of these learning difficulties and how the demonstration overcomes them would help convince other readers to try this method, especially if they are encountering the same issues with their students.*
   One major difficulty we observed in the exams of years before was a general lack of understanding of the UH-principle. Namely, that the unit hydrograph represents a transfer function that describes the "generation" of a hydrograph given a unit of effective precipitation in a catchment. We first included (for the case of a uniform distribution of effective precipitation) an interpretation of the UH as a distribution of travel times (similar to time-area histograms). It seemed not sufficient as still a large number of students were not able to formulate the functioning and interpretation of an UH properly. Our impression is, that with the introduction of our experiment understanding has improved (also many students now use the experiment as an example to explain the UH). However, given the large variation of students' backgrounds and capabilities from year to year, I would not dare to express any direct and sole effect of the experiment quantitatively. → We will add some specification of concrete difficulties as done here, but otherwise I would link to our answer to referee#1's comment 3.

3. *The organization of the paper as well as the figures are of good quality; the only concern regarding the writing pertains to some awkward phrasing and some typographic errors (see technical corrections in supplement).*
   See below.

*Specific Comments*

P2, Line 14 – This section starts at a nice review for the UH; however the Zoch and Clark references here are inserted without much description and are vague. If you want to use them as using 'similar concepts', I would suggest you provide more details.

As the introduction into the UH is already quite long, we prefer to not extend the description, and rather to delete that sentence.

P2, Line 19-21 – You should cite what you say here

Given the infiltration process as one of the processes that control effective precipitation, and given the non-linearity in the soil hydraulic properties, we do think it is obvious that this process is non-linear. We believe that additional explanation are not necessary in the text and also do not think it is necessary to have a citation for this fact. Nevertheless, we are happy to provide a reference, if required by the editor.

P2, Line 25 – Different conditions such as?

We will add them. Examples are: soil moisture, climate.

P2, Line 33 – I believe that the MHM model in the Samaniego paper does *not* explicitly use the UH concept as a routing method as it summarizes different grid cells through the regionalization process, then upscales to larger spatial scales and is not necessarily constrained to the UH assumptions/limitations. Perhaps double check the use of this citation.

The mHM uses a simple triangular UH to convolute runoff that is produced in each cell to represent spatially variable runoff production. The routing between cells in the catchment however is done differently. Therefore we would like to keep this citation of a current mesoscale hydrological model making use of the UH concept. → I suggest we add a sentence to address the use of the UH in the mHM model in a more concrete way.

P3, Line 12-13 – Could you elaborate a bit more here? Understanding how the students struggle here would help the reader understand how the activity improves their understanding

See answer to comment 1

Figure 1 – The figure illustrates the concept fairly well. I would choose colours/patterns that contrast more. Also make sure that the final version does not have blurry text

Will be accordingly changed.

*Technical Corrections*

P1, Line 12 – Unit-hydrograph and unit hydrograph are used interchangeably throughout the text and title. You should pick one and be consistent with it

P1, Line 13 – 'up-to-date' is awkward, perhaps use 'to date' or 'to this day'

P1, Line 15 – topic addressed in most of the (engineering) hydrology…

P1, Line 19 - experiment involving an active student….

P2, Line 10 – A step further has been the first attempt of a spatially distributed…First section is awkwardly phrased.

P2, Line 18 – Check that your references don't have the brackets {} for next submission. They also appear later on.

P2, Line 28 – Principal idea, not principle

P2, Line 30 – In other words,

P2, Line 36 -in any of the academic hydrology courses at the (BSc and MSc); I would say undergraduate and graduate level as they can be different in institutions or countries, e.g. BASc, B.Eng., etc.

P4, Line 1 – We would like to point out here, that the UH…

P4, Line 3 – a spatially explicit

P4, Line 8 - Units for runoff are given normalized by …

P4, Line 14 – 90 min

P5, Line 21 – The sampling of the water packages and along is carried out with…

P7, Line 16 - …by a yellow ball (first event from Figure 1)

P7, Line 20 - many of the recent

P7, Line 29 – (e.g. steep areas close to …. You need to close the parenthesis somewhere

P9, Line 12 – Confucius

Technical Corrections suggested by Ref. #2 will all be considered and implemented.

---

## Author Response (AR1)

**Responses to the Reviewer Comments and Description of Manuscript Changes**

(Referee comments in black; Responses in blue; Changes in the manuscript in red (page/line number refer to the marked-up-document))

First, we would like to thank both reviewers for their fair and very valuable comments. In the following, we have addressed each reviewer comments in detail and have indicated how we have altered and updated the manuscript accordingly.

We hope that we have addressed all comments sufficiently, and we are looking forward to your feedback and your decision.

When giving details to changes and updates in the manuscript the first page and line numbers refer to the submitted document, the second to the updated version.

**Comments by Referee #1:**

1. *This paper presents a method for interactively teaching students about the unit hydrograph. The approach taken is simple, involving the students passing balls along defined flow pathways so that the result at the "catchment outlet" can be observed. It is a simple, low-cost method of demonstrating a simple case of "unit hydrograph".*

   Reply: We thank the referee for this statement, as it was exactly our intension to develop such a simple, easy to implement and cheap experiment suitable for demonstration within a lecture.

   Changes: - None -

2. *Given the time needed to run each "experiment", I feel that a hybrid approach would be better, where the idea is introduced using a simple participatory demonstration as described here, but more detailed experiments are done through computer simulation. This is particularly the case when the time needed for a single experiment (including discussion) is between 30 and 90 minutes (page 9, line 4-5). 90 minutes is a considerable break in a 3 hour lecture, and suggests a more efficient method might be needed.*

   Reply a): We fully agree with the reviewer that it is also necessary to have additional exercises (e.g. in the computer lab), where students do explicit calculations applying the unit-hydrograph concept. We also do that in our program within the course "Exercises in Hydrology". The course is conducted separately, but organized in close cooperation. The lecture theatre experiment (as introduced in the paper) is a visceral aid that gives an additional visualization and participatory demonstration. The main goal is therefore to stimulate student interests and to help them in their scientific learning. The positive effect of experimental demonstrations on deeper understanding and learning has been found by a number of authors (e.g. Roberts et al., 2005, Savec et al., 2005).

   Changes a): We have include a separate paragraph addressing this topic and referring to these references in the revised version of the manuscript (page 3, line 16-21).

   Reply b): Concerning the time of the experiment – we wrote 30-90min in the submitted manuscript –, we could repeat the experiment and, with the help of 2 student assistants, we were able to include the basic demonstration within a 15-20min time slot. When this time slot is well set, it is an ideal interruption of a 3h lecture.

In the manuscript, we additionally describe variants of the experiment that can, but do not need to be performed. These variants are only described in case there is more time available.

Changes b): We have made these points clearer in the revised version of the manuscript (page 9, line 22-24).

3. *The real question here is: how many such experiments are needed in order to provide a suitable improvement in student understanding? Can a combination of participatory and computer examples achieve the same effect in less time?*

This is a very interesting question and depending on some funding, there might be a good chance over the next years to tackle this question. In general, a longer-term educational experiment would be required to answer this question thoroughly. The experiment will use some kind of split group approach and then analyze the exam/learning results for these different settings. As we have received the teaching award of our university for this experiment in the last year, our educational department had approached us in order to discuss such a long-term study to examine the effect. We apologize to admit that currently an answer to this question is out of scope.

Changes: In order to address this issue, we have slightly extended our discussion and added a paragraph on the need for such a longer-term investigation in the discussion (page 10, line 13-17), we also added a short paragraph giving some literature review on the effect of experimental demonstrations on the learning process (page 3, lines 24 - 27).

4. *The paper gives a reasonable review of the history of the unit hydrograph. I consider that the authors are incorrect in saying that the effective rainfall is homogeneously distributed over the catchment (page 4, lines 2-3). This is not necessarily the case. What the UH concept considers is that the spatial distribution of effective rainfall doesn't change between events. It can be non-homogeneously distributed. This can be due to spatial variations in rainfall (e.g. due to topographic effects), or due to spatial variations in the fraction of rainfall that is converted into effective rainfall (e.g. due to topography, soils, vegetation). Considering the effective rainfall to be homogeneously distributed across the catchment is a simple case, but not really the requirement of the unit hydrograph concept.*

We fully agree on this comment. The assumption of a uniform "effective precipitation" however is very often made in many textbooks as a requirement (e.g. Maniak, 2016, p350).

Changes: We deleted our statement about the requirement of homogeneous precipitation (page 4, line 4-5) and we added a sentence on the spatial distribution of precipitation at page 4, line 16-18.

5. *I think papers like this do have a place in HESS - but this paper needs a little more work in order to be of publishable quality.*

We hope our comments and suggested adaptations will sufficiently address the comments made by Referee #1.

Literature:

Maniak, U.: Hydrologie und Wasserwirtschaft, eine Einführung für Ingenieure, 7.Aufl., Springer Vieweg, 2016

Roberts, J. R., Hagedorn, E., Dillenburg, P., Patrick, M., and Herman,T.: Physical models enhance molecular three-dimensional literacy in an introductory biochemistry course, Biochem. Mol. Biol. Edu., 33, 105–110, 2005.

Savec, V. F., Vrtacnik, M., and Gilbert, J. K.: Evaluating the educational value of molecular structure representations, in: Visualisation in Science Education, edited by: Gilbert, J. K., Springer, 269–300, 2005.

**Comments by Referee #2**

1. *This paper presents a creative method of teaching the Unit Hydrograph, a fundamental concept in hydrology, in an interactive and non-lecture format. This would be a good addition to HESS as more and more instructors hope to incorporate nontraditional and active methods of teaching STEM concepts to suit different types of learning styles. While the demonstration may not be feasible in some cases, this paper presents one option of teaching the UH concept and could be the basis of different modifications to suit individual classroom needs.*
   We thank the referee for this statement!

2. *One general concern I had was that the authors cited "strong difficulties in students' perceptions of the UH' as the motivation for using the active demonstration. However, I could not find what specifically the previous concerns were and if they were actually addressed/reflected in the final evaluations after the demonstration. I believe summarizing some of these learning difficulties and how the demonstration overcomes them would help convince other readers to try this method, especially if they are encountering the same issues with their students.*
   One major difficulty we observed in the exams of years before was a general lack of understanding of the UH-principles. Namely, that the unit hydrograph represents a transfer function that describes the "generation" of a hydrograph given a unit of effective precipitation in a catchment. We first included (for the case of a uniform distribution of effective precipitation) an interpretation of the UH as a distribution of travel times (similar to time-area histograms). It seemed not sufficient as still a large number of students were not able to formulate the functioning and interpretation of an UH properly. Our impression was, that with the introduction of our experiment understanding has improved (also many students now use the experiment as an example to explain the UH). However, given the large variation of students' backgrounds and capabilities from year to year, I would not dare to express any direct and sole effect of the experiment quantitatively.
   Change: We added a paragraph to detail some of the difficulties (page 3, lines 16-21), and extended the conclusions as described under Rev.#1, comment 3.

3. *The organization of the paper as well as the figures are of good quality; the only concern regarding the writing pertains to some awkward phrasing and some typographic errors (see technical corrections in supplement).*
   See below.
   Change: - see specific comments -

*Specific Comments*

P2, Line 14 – This section starts at a nice review for the UH; however the Zoch and Clark references here are inserted without much description and are vague. If you want to use them as using 'similar concepts', I would suggest you provide more details.

As the introduction into the UH is already quite long, we prefer to not extend the description, and rather to delete that sentence.

Change: we deleted that that statement, in order to not extend the intro further and deleted the two citations from the list of references.

P2, Line 19-21 – You should cite what you say here

Given the infiltration process as one of the processes that control effective precipitation, and given the non-linearity in the soil hydraulic properties, we do think it is obvious that this process is non-linear. We believe that additional explanation are not necessary in the text and also do not think it is necessary to have a citation for this fact. Nevertheless, we are happy to provide a reference, if required by the editor.

Change: - None -

P2, Line 25 – Different conditions such as?

We will add them. Examples are: soil moisture, climate.

Change: we added (soil moisture conditions and climate) as explicit examples for different conditions (page 2, line 26).

P2, Line 33 – I believe that the MHM model in the Samaniego paper does *not* explicitly use the UH concept as a routing method as it summarizes different grid cells through the regionalization process, then upscales to larger spatial scales and is not necessarily constrained to the UH assumptions/limitations. Perhaps double check the use of this citation.

The mHM uses a simple triangular UH to convolute runoff that is produced in each cell to represent spatially variable runoff production. The routing between cells in the catchment however is done differently. Therefore we would like to keep this citation of a current mesoscale hydrological model making use of the UH concept.

Change: We explicitly stated the concrete use of the UH concept within mHM. We also deleted the reference Michel et al. 2003 as it primarily referring to an analysis of storage related formulations of catchment behavior.

P3, Line 12-13 – Could you elaborate a bit more here? Understanding how the students struggle here would help the reader understand how the activity improves their understanding

See answer to comment 1

Change: as described under comment 1 (page 3, lines 16-21)

Figure 1 – The figure illustrates the concept fairly well. I would choose colours/patterns that contrast more. Also make sure that the final version does not have blurry text

Change: Figure 1 has been adapted so that pattern contrast more.

*Technical Corrections*

P1, Line 12 – Unit-hydrograph and unit hydrograph are used interchangeably throughout the text and title. You should pick one and be consistent with it (consistently changed to "unit hydrograph")

P1, Line 13 – 'up-to-date' is awkward, perhaps use 'to date' or 'to this day' (changed to "to this day")

P1, Line 15 – topic addressed in most of the (engineering) hydrology…(rephrased)

P1, Line 19 - experiment involving an active student….(rephrased)

P2, Line 10 – A step further has been the first attempt of a spatially distributed…First section is awkwardly phrased. (rephrased)

P2, Line 18 – Check that your references don't have the brackets {} for next submission. They also appear later on. (changed to "( . )" throughout the whole manuscript.)

P2, Line 28 – Principal idea, not principle (changed)

P2, Line 30 – In other words, (changed)

P2, Line 36 -in any of the academic hydrology courses at the (BSc and MSc); I would say undergraduate and graduate level as they can be different in institutions or countries, e.g. BASc, B.Eng., etc. (changed accordingly.)

P4, Line 1 – We would like to point out here, that the UH… (changed)

P4, Line 3 – a spatially explicit (changed)

P4, Line 8 - Units for runoff are given normalized by …(changed to "Units for runoff are provided as catchment area specific runoff in …"

P4, Line 14 – 90 min (changed)

P5, Line 21 – The sampling of the water packages and along is carried out with… (corrected to "The sampling of the water packages (balls) and along the "main stream channel" is carried out …")

P7, Line 16 - …by a yellow ball (first event from Figure 1) (Captions for figure 1 are rephrased)

P7, Line 20 - many of the recent (changed to "While such an analysis is presented in recent hydrology textbooks .." )

P7, Line 29 – (e.g. steep areas close to …. You need to close the parenthesis somewhere (done)

P9, Line 12 – Confucius (sorry we used the German spelling; we changed the name to Confucius)

[revised manuscript text omitted]